# The Improved Cytotoxic Capacity of Functionalized Nanodiamonds with Metformin in Breast and Ovarian Cancer Cell Lines

Lucero Evelia Acuña-Aguilar [1] , Alain Salvador Conejo-Dávila [1,2], Mario Miki-Yoshida [1],
Olga N. Hernández-de la Cruz [3], Gricelda Sánchez-Sánchez [3], César López-Camarillo [3] ,
Joan Sebastian Salas-Leiva [1,4] , Erasto Armando Zaragoza-Contreras [1] , Reyna Reyes-Martínez [5]
and Erasmo Orrantia-Borunda [1,*]

1    Departamento de Medio Ambiente y Energía, Centro de Investigación en Materiales Avanzados,
     S.C. Av. Miguel de Cervantes Saavedra 120, Chihuahua 31136, Mexico;
     lucero.acuna@cimav.edu.mx (L.E.A.-A.); alain.conejo@cimav.edu.mx (A.S.C.-D.);
     mario.miki@cimav.edu.mx (M.M.-Y.); joan.salas@cimav.edu.mx (J.S.S.-L.);
     armando.zaragoza@cimav.edu.mx (E.A.Z.-C.)
2    Centro de Innovación Aplicada en Tecnologías Competitivas, A.C. Calle Omega No. 201, Delta,
     León 37545, Mexico
3    Posgrado en Ciencias Genómicas, Universidad Autónoma de la Ciudad de México,
     Ciudad de México 06720, Mexico; ediacara79@yahoo.com.mx (O.N.H.-d.l.C.);
     gricelda.sanchez.sanchez@estudiante.uacm.edu.mx (G.S.-S.); cesar.lopez@uacm.edu.mx (C.L.-C.)
4    CONAHCyT-Departamento de Medio Ambiente y Energía, Centro de Investigación en Materiales Avanzados,
     S.C. Av. Miguel de Cervantes 120, Chihuahua 31136, Mexico
5    Departamento de Química, Facultad de Ciencias Químicas, Universidad Autónoma de Chihuahua,
     Ciudad Universitaria, Campus 2, Chihuahua 31110, Mexico; rreyesm@uach.mx
*    Correspondence: erasmo.orrantia@cimav.edu.mx; Tel.: +614-439-1111 (ext. 2412)

**Abstract:** Nanodiamonds (ND-COOH) are used as drug delivery systems because of their attractive properties, as they allow for optimized transport of therapeutic agents in cellular models. Metformin (MET) is a drug used in diabetes mellitus therapy and exhibits anti-cancer properties. In this study, dispersed nanodiamonds were functionalized with metformin by directly binding them to 1,6-hexanediol (ND-MET), and their effects on the cytotoxicity of breast and ovarian cancer cells were evaluated in vitro. A simple synthesis of ND-MET was performed and characterized using FT-IR, XPS, Boehm titration, RAMAN, XDR, TEM, and dynamic light scattering (DLS). Data showed an increased intensity of the C-N bond band, indicating the presence of metformin in ND-MET. We detected signals at 1427 $cm^{-1}$ and 1288 $cm^{-1}$ corresponding to the C-N and C-H bonds, and adsorptions at 1061 $cm^{-1}$ and 3208 $cm^{-1}$ corresponding to the N-O and N-H bonds, respectively. The deconvolution of the C1s binding energy was also found at 286.24 eV. The biological effects of ND-MET were tested in both SKOV3 ovarian cancer and Hs-578T and MDA-MB-231 triple-negative breast cancer cell lines. In SKOV3 cells, the $IC_{50}$ for the ND-MET complex was $35 \pm 14$ µg/mL, while for Hs-578T and MDA-MB-231 breast cancer cells, the $IC_{50}$ for ND-MET was $759 \pm 44$ µg/mL and $454 \pm 49$ µg/mL, respectively. Our data showed that ND-MET could be used as an intracellular delivery system for metformin in cancer cells. Cell viability assays evidenced a reduced viability of all cell lines in a time and dose-dependent manner, with a significant sensitivity observed in SKOV3 ovarian cancer cells treated with ND-MET.

**Keywords:** nanodiamonds; biguanide; functionalization; ovarian cancer; breast cancer

## 1. Introduction

Nanodiamonds (ND-COOH) are extensively used as drug delivery systems. Due to their chemical and physical surface properties, nanodiamonds allow for optimized

delivery of various therapeutic agents [1–4], and can diffuse within cells due to their lipophilicity [5–7]. In addition, longer drug retention has been observed when associated with nanodiamonds compared to the drug alone. Since metformin (MET) can be conjugated via ionic bonds, covalent bonds, or physical adsorption to the nanodiamond, it is possible to have a prolonged and controlled pH-mediated release. This conjugation minimizes anti-cancer drug side effects and maximizes delivery efficiency [8].

MET is a drug used in treating type II diabetes since it inhibits gluconeogenesis [9], and also seems to exert a cytotoxic effect on cancer cells. Its potential use in anti-cancer processes is currently being investigated [10–16]. The effect of MET on primary breast cancer cells (PBCC) has been studied, and assays show a dose-dependent reduction of viability and proliferation in ER+, PR+, and HER2- type cancers. MET, at an effective dose (25 mM, 24 h), can inhibit tumor cell proliferation and metastasis by modulating the expression of metalloproteinases MMP-2 and MMP-9, and interfering with NF-kB protein signaling in PBCC [17]. Also, MET has been delivered to graphene oxide nanoparticles, loaded with hyaluronic acid (HA-GO-MET), and targeted to both CD44 receptors and overexpressing cancer cells, with MET as the therapeutic agent and hyaluronic acid (HA) as the active targeting system. Results indicated an inhibition of cell migration of MDA-MB-231, a triple-negative breast cancer (TNBC) line, when treated with significantly lower doses of HA-GO-MET (20 μg/mL) compared to metformin alone (300 μg/mL); it attained 50% viability. The GO-MET system evaluated at a 50 μg/mL dose, and showed no significant inhibition in cell viability [18].

MET requires high doses to inhibit cancer cell growth depending on the type and subtype of cell line [19], the metformin derivative [12], and the duration of treatment being evaluated. For example, the $IC_{50}$ at 48 h of MET treatment for SKOV3 was 14.92 mM [16], and for Hs578t and MDA-MB-231, it was 16.3 mM and 51.4 mM, respectively [19]. This high dose is a disadvantage; however, it can be overcome using efficient drug delivery systems that selectively transport MET to the target cells [20]. Therefore, functionalizing nanodiamonds with MET is an interesting way to evaluate their effect on cancer cell lines [21,22].

Since we are interested in the search for a compound that can serve as a treatment against tumor cells, we decided to work with two types of cancer that are difficult to treat because they present resistance to conventional therapies. We looked at triple-negative breast cancer and serous ovarian cancer subtypes. Triple-negative breast cancer is resistant to hormonal therapies and serious breast cancer to chemical therapies. Considering this, we chose representative and widely used cell lines to study these tumor subtypes, MDA and HS (triple negative), and SKOV (serous).

The resistance of tumor cells to specific therapies represents a significant challenge in cancer treatment. In breast cancer, the subtype that typically exhibits a lower response to therapies and has a poorer prognosis is triple-negative breast cancer (TNBC) [23]. TNBC is characterized by the absence of specific molecular markers on the cell surface: estrogen, progesterone, and HER2/neu receptors. These markers are crucial in targeted therapy against other breast cancer subtypes. The lack of these receptors in TNBC renders this type of cancer unresponsive to hormone-targeted therapies, such as tamoxifen, which are effective in tumors expressing estrogen or progesterone receptors. Moreover, due to the lack of HER2/neu receptor expression, triple-negative breast cancer does not respond to HER2-targeted therapies like trastuzumab [24–26].

Consequently, the inefficacy of these specific therapeutic options makes TNBC more challenging to treat when compared to other subtypes of breast cancer. Hence, we decided to evaluate nano-compounds' effects on the viability of TNBC tumor cells (MDA-MB-231 and Hs578t). Additionally, TNBC is highly aggressive, and cell lines of this subtype serve as excellent models for studying invasion and metastasis processes, making them relevant models for advanced-stage cancer.

In ovarian cancer (OC), the serous subtype is highly prevalent. It can exhibit high aggressiveness and be associated with a poor prognosis and reduced responsiveness to

therapies compared to other OC subtypes. SKOV3 is a cell line derived from serous ovarian carcinoma. It has been extensively employed as a study model due to its resistance to certain chemical compounds traditionally used for OC treatment, such as Paclitaxel and Adriamycin [27,28].

This study evaluated the cytotoxic capacity of nanodiamonds (ND-COOH) modified with MET in different cancer cell lines. The objective was to verify whether the action of ND-MET can reduce the amount of MET necessary to induce appreciable biological effects. Moreover, the functionalization of nanodiamonds with metformin was achieved using a straightforward method.

## 2. Materials and Methods

### 2.1. Regents

Carboxylated nanodiamonds (ND-COOH) monodispersed in water with a concentration of 10 mg mL$^{-1}$ and a particle size of 5–6 nm were obtained from Sigma-Aldrich (Milwaukee, WI, USA). 1,6-hexanediol ($C_6H_{14}O_2$), N, N-dimethylformamide (DMF), water ($H_2O$), and sodium hydroxide (NaOH) were obtained from MACRON Fine Chemicals (Xalostoc, Estado de México, Mexico). Acetone ($CH_3COCH_3$) was obtained from Química Tech SA de CV (MX). Ethanol (EtOH) was obtained from CTR Scientific (MX). Sulfuric acid ($H_2SO_4$) and potassium permanganate ($KMnO_4$) were obtained from Fermont (MX). Hydrochloric acid (HCl) was obtained from JT Baker (Phillipsburg, NJ, USA), and the Metformin 850 mg tablet was obtained from the Pisa pharmaceutical company (San Diego, CA, USA).

### 2.2. ND-MET Synthesis

#### 2.2.1. Metformin Hydrochloride Extraction (MET*HCl)

Eight metformin tablets, each containing 850 mg, were triturated and added to an Erlenmeyer flask with 400 mL of ethanol, forming a suspension, and left in constant agitation at 25 °C for 2 h. It was filtered to separate the solids, then the solution was recovered and concentrated at a constant pressure until a white crystalline solid was obtained. Metformin was extracted by the Universidad Autónoma de Chihuahua Department of Chemistry.

#### 2.2.2. Obtention of ND-MET Complex

The functionalization of the nanodiamonds started with a Fisher esterification reaction using 1,6-hexanediol and ND-COOH in a 1:10 mass ratio. In total, 2 mL of N, N-dimethylformamide (DMF) and 20 μL of $H_2SO_4$ were added. The mixture was placed in an oil bath at 100 °C and stirred for 6 h. It was then cooled to 25 °C, centrifuged at 900 rpm, washed twice with deionized water, and once with acetone. The dispersion was sonicated between each wash. Then, the solvent was evaporated at 25 °C to obtain ND$_{hexa}$ (Figure 1a). Liu et al. functionalized nanodiamonds with 1,6-hexanediol using $SOCl_2$ to bind them to biopolymers [29] covalently.

The functionalization of the nanodiamonds was continued with the oxidation of the primary alcohols from 1,6-hexanediol to obtain the carboxylic functional groups (-COOH) [30]. A solution of NaOH 1 N was mixed with 100 mg of ND$_{HEXA}$. Afterward, a solution of $KMnO_4$ 0.17 equivalents relative to 1,6-hexanediol was prepared and added dropwise to the previous solution in an ice bath, then stirred for 30 min. The ice bath was removed to stir for 18 h. The pH was adjusted to 3–4 with HCl 0.1 N, centrifuged at 900 rpm, washed with deionized water with 5 min sonication between each wash, and dried at 25 °C to obtain the ND$_{HEXA}$-$CO_2$H, as shown in Figure 1b.

MET was linked to ND$_{HEXA}$-$CO_2$H through a neutralization reaction between the carboxylic acids group (which has a pKa of 4.75) and the MET's amino group (pKa of 10.73). The reaction produces electrostatic interactions between the drug and the nanodiamond, which allows its transport and easy delivery. For drug binding, 50 mg of ND$_{HEXA}$-$CO_2$H was mixed with 50 mg of metformin on agitation for 12 h at 25 °C, washed with deionized water, then dried at room temperature to obtain ND-MET.

**Figure 1.** Reaction sequence for the synthesis of the ND-MET complex, (**a**) 1:10 ND-COOH:1,6-hexanediol, DMF, and $H_2SO_4$, (**b**) 1N NaOH, $KMnO_4$ and HCl, (**c**) Metformin.

*2.3. Characterization*

The characterizations of ND-MET complex synthesis were performed by FT-IR, XDR, XPS, and RAMAN. The particle size and shape help determine nanoparticle chemical and physical stability.

2.3.1. Fourier Transform Infrared Spectroscopy

Functional groups were identified using an FT-IR spectrometer (IR Affinity 1S, Shimadzu, Kyoto, Japan). Spectra were obtained by reflectance using a Smiths ATR (Total Attenuated Reflectance) accessory, model Quest, with a one-step diamond window.

2.3.2. X-ray Photoelectron Spectroscopy

X-ray photoelectron spectroscopy (XPS) analysis was performed on the Escalab 250Xi (ThermoFisher, Waltham, MA, USA) to confirm the presence of carboxyl groups. An Al K alpha source, with a 650 μm spot size, was used in CAE analysis mode with a pass energy of 200 eV and an energy step size of 1.000 eV.

2.3.3. Bohem Titration

The Bohem titration was performed with an automatic titrator Metrohm model 848 Titrino Plus. An HCl 0.1 M solution was prepared to titrate the $NaHCO_3$ 0.05 M base containing the sample and quantify the carboxyl groups of ND-COOH.

2.3.4. RAMAN Spectroscopy

A RAMAN spectrometer (LabRamHR Vis-633, Horiba, Kyoto, Japan) complemented functional group characterization. The sample was excited to analyze the molecular structure with a laser λ = 325 nm.

### 2.3.5. X-ray Diffraction Analysis

X-ray diffraction (XRD) was run on the X'PertPRO (PANalytical, Boston, MA, USA) diffractometer using CuKα radiation with λ = 1.54056 Å in the scanning range 2θ = 10°–100°, with a step of 0.01 and a time of 100 s/step, with Bragg Brentano. The analysis was performed to identify the nanodiamond crystalline structure and crystallite size.

### 2.3.6. Transmission Electron Microscopy

Transmission electron microscopy (TEM) with an energy dispersive system (EDS) was performed on JEM 2200FS+CS (JEOL, Ciudad de México, MX) equipment, with a spherical aberration corrector of the condenser lens for the scanning transmission mode (STEM), operated at 200 kV. With this technique, the nanoparticles' morphology and particle sizes were obtained.

### 2.3.7. Dynamic Light Scattering

The hydrodynamic radius and Z-potential measurements were carried out with the Zetasizer nano series (Malvern Panalytical's, Worcestershire, UK) to analyze the particle size and charge. Nine measurements were performed using distilled water as the solution medium.

### *2.4. Biological Assays*
### 2.4.1. Cell Culture

SKOV3 ovarian cancer cells, MDA-MB-231, and Hs578t breast cancer cells were purchased from ATCC (Manassas, VA, USA) and maintained in a DMEM/F-12 (Dulbecco's Modified Eagle Medium/Nutrient Mixture F-12; Gibco, BRL, Grand Island, NY, USA) medium supplemented with 10% fetal bovine serum (FBS; Gibco BRL, Grand Island, NY, USA) and 1% penicillin/streptomycin antibiotics (Gibco BRL, Grand Island, NY, USA). The cells were cultured in a humidified atmosphere with 5% $CO_2$ at 37 °C and subcultured every 2–3 days.

### 2.4.2. Cytotoxicity Assay

The cytotoxic capacity of the ND-MET bioconjugates on the breast and ovarian cancer cell lines was evaluated using a 3-(4,5-dimethylthiazol-2-yl)-2,5-diphenyltetrazolium bromide (MTT) colorimetric assay. The different cell lines were cultured in vitro in 96-well plates (10,000 cells per well) until they reached 70–80% confluency. Subsequently, the cells were treated with different concentrations (15, 30, 60, 125, 250, 500, and 750 μg/mL) of the ND-MET nanoparticles, which were then dissolved in a DMEM-F12 complete medium. Afterward, they were incubated at different times (24, 48, and 72 h) at 37 °C and 5% $CO_2$. Cells without treatment were used as an experimental control. When post-treatment times were complete, cell viability was analyzed using MTT assays. Thus, the medium with the treatment was removed and replaced with 100 μL of an MTT solution (0.5 μg/mL in DMEM-F12 medium), which was then incubated for 4 h at 37 °C and 5% $CO_2$. After the medium with MTT was aspirated, 100 μL of extraction buffer (0.1 N HCl dissolved in isopropanol) was added to each well to dissolve the resulting purple formazan crystals. The absorbance of the resulting solution was measured at 570 nm in a microplate reader (Elx800-BioTek Instruments). A plot of absorbance versus concentration was generated to determine the $IC_{50}$ value using Quest Graph TM $IC_{50}$ 2023 software (ATT Bioquest) [31]. The values obtained from the control cells, without treatment, were considered 100% viable and were used to calculate the viability of the tumor cells treated with ND-MET nanoparticles.

For IC50 data, a one-way analysis of variance (ANOVA) and a Turkey post-hoc test were conducted to investigate differences in the IC50 across three treatment groups [32].

## 3. Results and Discussion

### 3.1. ND-MET Characterization

Figure 1 illustrates the reaction sequence to obtain the ND-MET complex, where an electrostatic interaction occurs between the carboxylic group of the $ND_{HEXA}\text{-}CO_2H$ reaction intermediate and the amino groups of the MET's molecule, since an attractive force is occurring between two opposite ionic species.

The FT-IR spectrum was used to identify the functional groups of the nanodiamond particles (ND-COOH), the products of the different functionalization steps, and the ND-MET complex. Figure 2 shows the transmission bands of ND-COOH attributed to the stretching bond of the hydroxyl group at 3387 cm$^{-1}$, the stretching of the carbonyl group at 1763 cm$^{-1}$, the bending of the hydroxyl bond at 1630 cm$^{-1}$, and the stretching bond of C-OH at 1245 cm$^{-1}$, which all confirm the presence of carboxylic acids in the structure of the nanodiamond. It is worth mentioning that nanodiamonds present various oxygen-functional groups on the surface, such as aldehydes, ketones, hydroxyls, and carboxylic acids, making it ideal for surface functionalization [33].

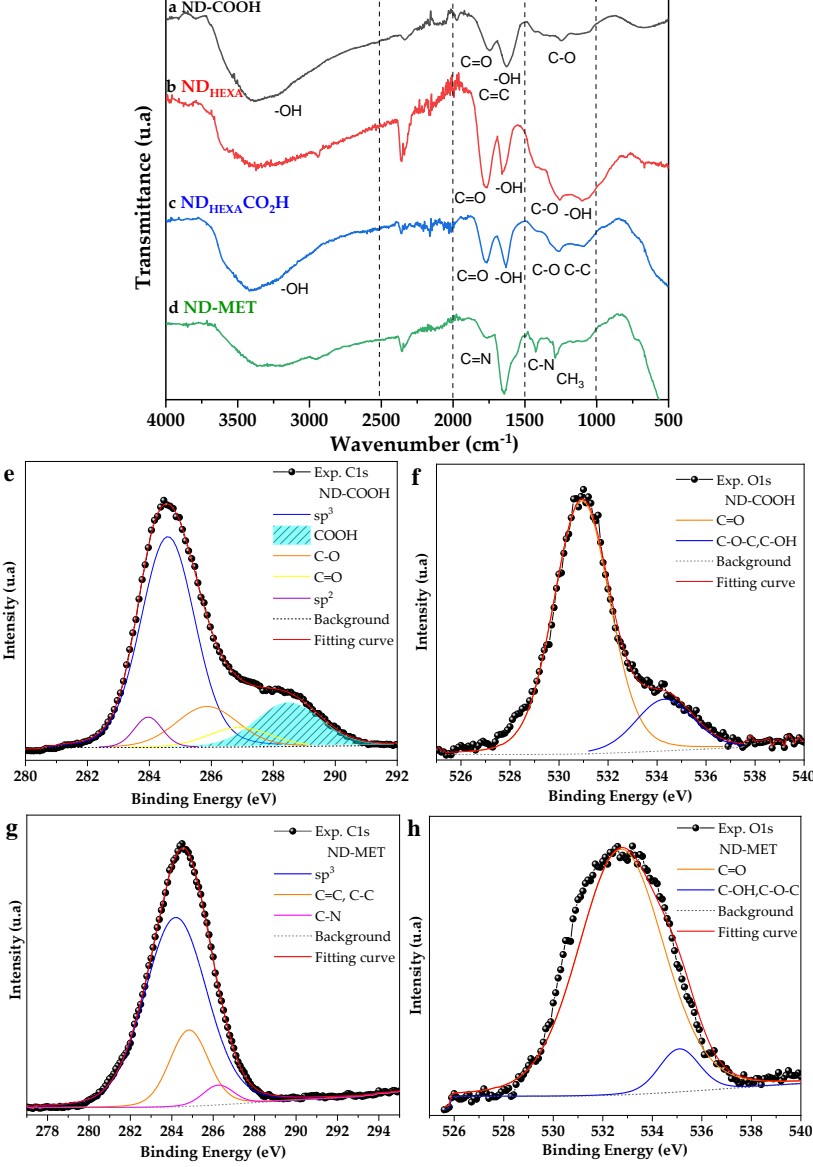

**Figure 2.** (**a–d**) show infrared spectra of nanodiamonds (ND-COOH) functionalized with MET, (**e**) deconvolution of C1s of nanodiamonds, (**f**) deconvolution of O1s of nanodiamonds, (**g**) deconvolution of C1s of ND-MET and (**h**) deconvolution of O1s of ND-MET.

When the nanodiamonds were esterified with 1,6-hexanediol, the stretching of the C-O bond of the ester formed can be observed at 1259 cm$^{-1}$, and the signal from the primary -OH of 1,6-hexanediol can be observed at 1075 cm$^{-1}$, as shown in Figure 2b. The surface oxidation reaction of the primary alcohols present in the nanodiamond to generate carboxylic acid groups (Figure 2c) was carried out using potassium permanganate (KMnO$_4$). The formation of the acid groups is confirmed by the presence of the signals at 1778 and 1632 cm$^{-1}$ corresponding to the stretching of the carbonyl group and the bending of the hydroxyl, respectively.

MET presents several primary amines (-NH$_2$) in its structure (Figure 1c). These functional groups show signals between 1640–1560 cm$^{-1}$ and 3500–3400 cm$^{-1}$. When metformin reacts with ND$_{HEXA}$CO$_2$H, the intensity of the signals coming from the primary amines decreases. In addition, a new signal appears at 2850 cm$^{-1}$ due to the quaternization of the nitrogen of the amine that reacted with the acid groups of the ND$_{HEXA}$CO$_2$H. Secondary amines appear at 1427 cm$^{-1}$. Moreover, the ND-MET spectrum (Figure 2d) presents the signals from the bond stretching of the carbonyl group that decreases in intensity at 1778 cm$^{-1}$. The two signals at 1427 and 1288 cm$^{-1}$ corresponding to the C-N and CH$_3$ bonds, respectively, confirm the formation of the ND-MET.

XPS analysis was performed to confirm the abundance of carboxylic groups and to determine the presence of bonds in ND-COOH and ND-MET. The C1s and O1s of both samples were deconvoluted using the Lorentzian fit. Figure 2e shows the deconvolution of the ND-COOH C1s peak into five binding energy bands at 288.48 eV (COOH), 286.89 eV (C=O), 285.86 eV (C-O), 284.60 eV (sp$^3$), and 283.60 eV (sp$^2$). In Figure 2f, the O1s peak was deconvoluted into two binding energies at 530.94 eV (C=O, OH-) and 534.32 eV (C-OH, C-O-C, H$_2$O).

It is possible to observe that the large area of the carbonyl deconvolution curve for the C1s and O1s of ND-COOH shown at 286.89 eV and 530.94 eV (C=O) binding energies, respectively, indicate a high density of the carbonyl functional group on the ND-COOH surface [33].

Figure 2g indicates the deconvolution of the C1s of the ND-MET complex at three binding energies: 284.19 eV (sp$^3$), 284.83 eV (C=C, C-C), and 286.24 eV (C-N). Figure 2h shows the deconvolution of the O1s of ND-MET at two binding energies: 532.09 eV (C=O) and 534.28 eV (C-OH, C-O-C, H$_2$O). Comparing Figure 2e,g, it is observed that the area of COOH at 288.49 eV no longer appears in the ND-MET sample, which could be attributed to the binding of metformin to this functional group. In addition, an atomic percentage of 3.96% belongs to the C-N bond, which presents a binding energy at 286.24 eV in Figure 2g.

Boehm titration was used to quantify the carboxylic groups on the ND-COOH surface. Table 1 shows the COOH sites for both the nanodiamond (ND-COOH) and the nanodiamond oxidized with KMnO$_4$, increasing from 1.10 ± 0.04 COOH per nm$^{-2}$ to 9.36 ± 1.42 COOH per nm$^{-2}$, respectively. The acid-oxidized nanodiamonds reported average carboxylic group values of 0.85 COOH per nm$^{-2}$ after treatment [34,35]. Using KMnO$_4$ to oxidize the primary alcohol to carboxylic acid proved more efficient for enhancing this functional group.

**Table 1.** COOH sites for ND-COOH and ND$_{HEXA}$-CO$_2$H.

| Sample | COOH Sites (COOH/nm$^2$) |
| --- | --- |
| ND-COOH | 1.10 ± 0.04 |
| ND$_{HEXA}$-CO$_2$H | 9.36 ± 1.42 |

RAMAN spectroscopy was used to complement molecular structure characterization. Figure 3a–e show the Raman spectra for ND-COOH, MET, and each of the derivatives obtained during the functionalization, indicating the changes in the molecule. ND-COOH exhibits stretching modes for C-Hx and O-H near 3000 and 3500 cm$^{-1}$, respectively. As seen in Figure 3a, around 1330 cm$^{-1}$, the nanodiamond shows an asymmetric peak, denoted as

the D-band, representing the sp$^3$ hybridization of the diamond. Amorphous carbonaceous compounds show a band denoted G in the range 1400–1800 cm$^{-1}$; this band does not appear when the sp$^3$ carbon form is pure. Some interpretations for this band are the existence of sp$^2$ carbon, an sp$^2$ cluster, an sp$^2$/sp$^3$ mixture, surface hydroxyl groups, and C=C as defects within the diamond structure [36,37]. Mochalin et al. assigned it to surface O-H bending vibrations or absorbed water. For the ND-MET complex in Figure 3d, two bands would indicate the binding of metformin to the nanodiamond; the vibrational mode at 1061 cm$^{-1}$ as ascribed to the N-O bond, which is the one that would be formed, and the presence of N-H at 3208 cm$^{-1}$, evidences the metformin molecule [38]. The RAMAN spectrum of MET in Figure 3e includes the C-N-C deformation at 480 cm$^{-1}$ and 724 cm$^{-1}$, the C=N stretching vibration at 1592 cm$^{-1}$, the methyl group -CH$_3$ stretching at 2951 cm$^{-1}$, the presence of nitrogen near the methyl group which reduces the CH$_3$ stretching symmetry, and at 3183 cm$^{-1}$, the N-H bond of the C=N-H stretching. This vibration frequency usually decreases in the presence of the hydrogen bond [39]. Other bands at 3300–3800 cm$^{-1}$ represent C-H and N-H [40,41]; the bands at 3355 cm$^{-1}$ and 3183 cm$^{-1}$ are specific for the asymmetric and symmetric N-H stretching vibrations, respectively [39].

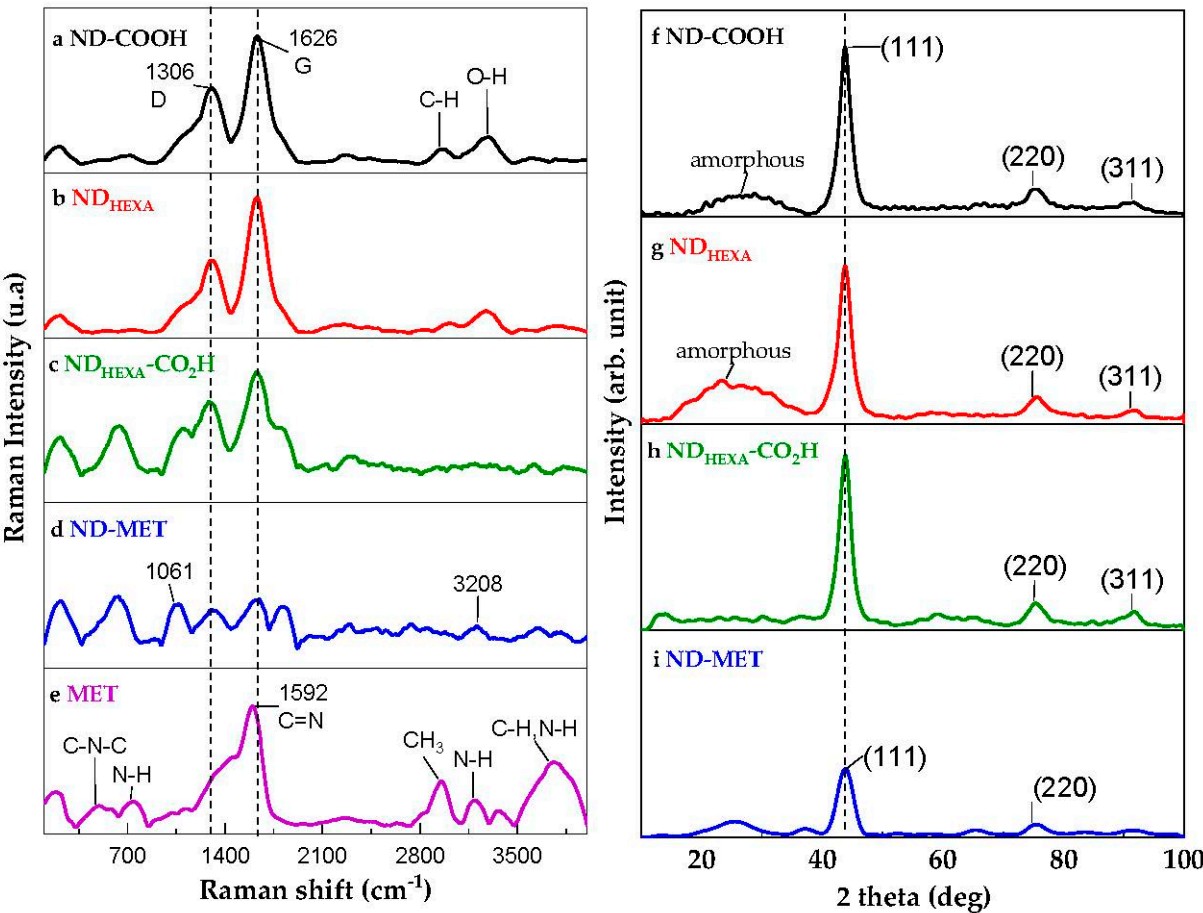

**Figure 3.** (**a**–**e**) Raman spectrum of ND-COOH, MET, and its derivatives and (**f**–**i**) XDR of ND-COOH and its derivatives from functionalization of ND-MET.

The crystallinity of ND-COOH and ND-MET was investigated using X-ray diffraction. The diffraction patterns are shown in Figure 3f–i. The XRD spectrum of ND-COOH (Figure 3f) indicates three dominant peaks at 2θ = 43.81°, 75.54°, and 91.81°, corresponding to the (111), (220), and (311) planes, respectively. These results evidence the crystallinity of ND-COOH, corresponding to a cubic structure (PDF ICDD 03-065-6329). In addition, the interplanar distance (*d*) measured on the high-resolution micrograph (Figure 4a), d = 2.10 ± 0.02, corresponds to the most intense line (111) of the XDR.

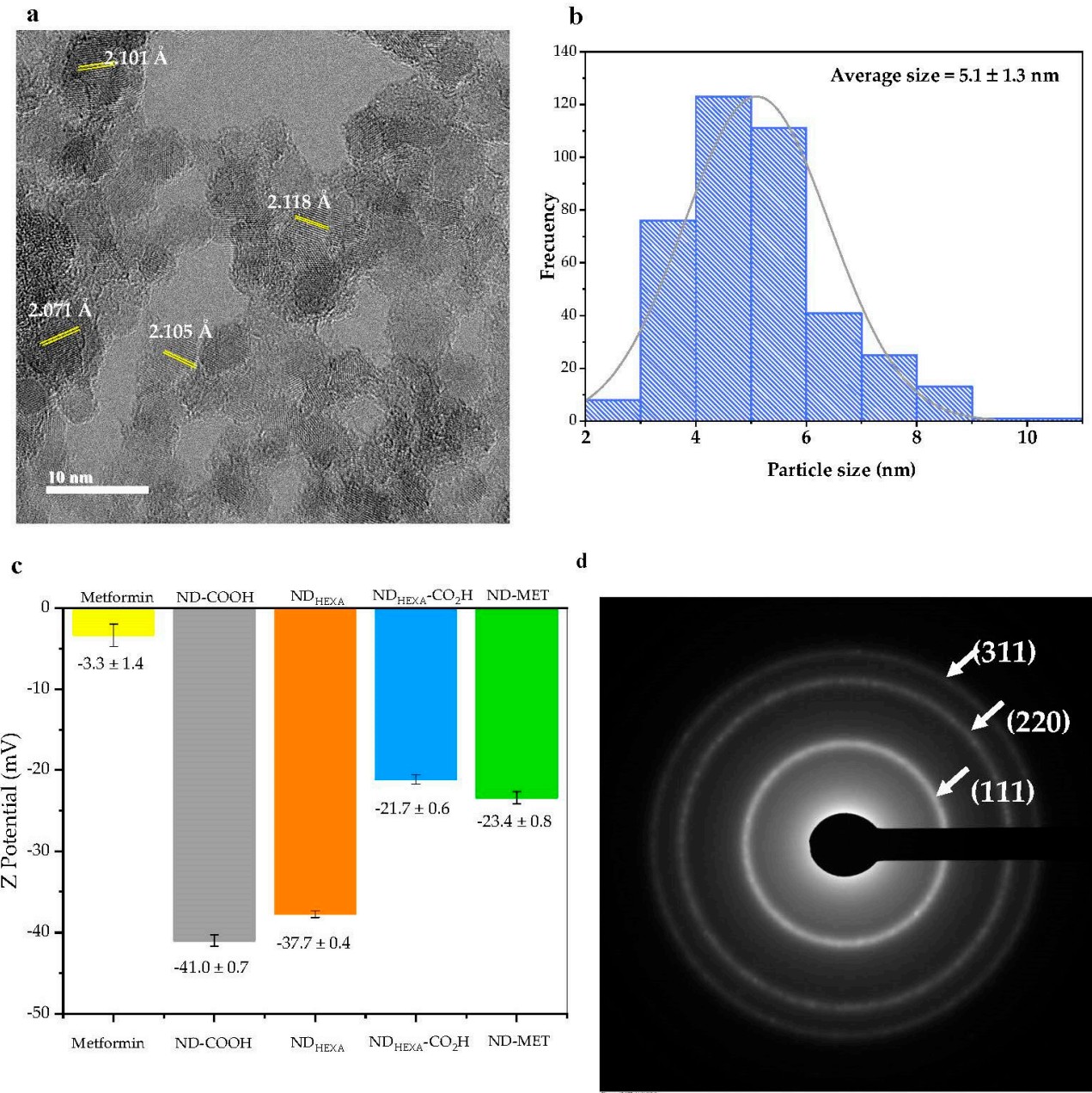

**Figure 4.** (**a**) High resolution micrograph of ND-MET, (**b**) particle size distribution of ND-MET, (**c**) Z-potential of ND-COOH, ND-MET, and its derivatives, and (**d**) diffraction pattern of ND-MET.

A band in the 20°–35° range is observed; this band is associated with amorphous carbon [42,43] because the crystalline nanodiamond's surface can undergo a phase transition during synthesis. This phase transition is also verified using XPS analysis in Figure 2e, where the area under the curve, corresponding to the $sp^2$ hybridization of the nanodiamond, is shown. In Figure 3g, the intensity of the amorphous band is increased by adding 1,6-hexanediol; this occurs because it is an amorphous organic compound that contributes to the increase of the intensity of the 20°–35° band. In $ND_{HEXA}CO_2H$, oxidizing the alcohol with $KMnO_4$ also oxidizes the surface of graphene, forming oxidized graphene (GO), which shows a peak around 10° [43], as seen in Figure 3h.

When the drug is added, the intensity of the (111) and (311) planes of the nanodiamond decreases, which indicates that the percentage of crystallinity decreases. For all derivatives, the peaks remain constant in the same position during the process, so it is possible to determine that the cubic phase of the nanodiamond is maintained.

High-resolution TEM, Figure 4a, presents the morphology of ND-MET. As noted, the material exhibits a globular morphology and interplanar distance of about 2.059 ± 0.020 Å, corresponding to the (111) plane of the nanodiamond. The sample shows a group of particles of uniform size, Figure 4b. The individual size of the ND-MET averages 5.1 ± 1.3 nm in diameter, similar to that obtained for ND-COOH, which was 5.3 ± 1.3 nm.

The particle size determines nanoparticles' biocompatibility, toxicity, and surface area. The charge and size of the nanoparticle also depends on their particle size distribution. Positively charged particles are retained in the liver, spleen, and lungs, while the neutral or slightly negatively charged ones show prolonged circulation in the blood [44].

Figure 4c shows the Z potential, obtained by DLS, of ND-COOH, ND-MET, and their functionalization derivatives obtained by DLS. The ND-COOH presented a hydrodynamic radius of 13.80 ± 0.04 nm (±SD) on average, and a Z potential of −41.0 ± 0.7 mV (±SD); this value indicates a stable system, free of agglomeration [45]. The degree of ionization of the carboxylic groups determines the strength of the interparticle interaction. When the carboxylic groups are ionized, the ND-COOH particles become negatively charged, and the repulsion between them becomes significant, resulting in a size decrease [33].

Figure 4c shows the decrease of the Z potential value for the functionalization derivatives ($ND_{HEXA}$ and $ND_{HEXA}COOH$). This change in the Z potential is attributed to the conversion of anionic carboxylate groups on the surface of the nanodiamond to charge the neutral 1,6-hexanediol, giving a network of decreasing negative charge on the particle's surface. The Z potential varies if the pH changes. The Z potential for pH 7 corresponds to −21 mV for the carboxylated nanodiamond. This change indicates the deprotonation of the -COOH group, causing the nanodiamond (ND-COOH) surface to become negatively charged [33], making it possible to obtain better functionalization.

On the other hand, Figure 4d displays the diffraction pattern of the ND-MET, showing the rings associated with interplanar distances 2.11 Å, 1.30 Å, and 1.10 Å, corresponding to the (111), (220), and (311) planes, respectively.

### 3.2. Cytotoxicity Evaluation of ND-MET in Breast and Ovarian Cancer Cell Lines

Antitumor properties of nanodiamonds (ND-COOH), drug MET alone, and ND-MET complex on ovarian cancer (SKOV3) and triple-negative breast cancer (Hs578t and MDA-MB-231) cell lines were evaluated using a MTT assay. The rationale for the use of these cancer cell lines have been documented in many publications. Hs578t and MDA-MB-231 breast cancer cells exhibit a lower response to therapies and have a poorer prognosis; they are an example of triple-negative breast cancer (TNBC). The lack of estrogen, progesterone, and HER2/neu receptors in TNBC renders this type of cancer unresponsive to hormone-targeted therapies, such as tamoxifen, which are effective in tumors expressing estrogen or progesterone receptors. Moreover, due to the lack of HER2/neu receptor expression, TNBC does not respond to HER2-targeted therapies, like trastuzumab. Consequently, the inefficacy of these specific therapeutic options makes TNBC more challenging to treat when compared to other subtypes of breast cancer. Hence, we decided to evaluate the effect of nano-compounds as a novel therapeutic tool. For ovarian cancer (OC), the serous subtype is highly prevalent and can exhibit high aggressiveness, and is also associated with a poor prognosis and reduced responsiveness to therapies. SKOV3 is a cell line derived from serous ovarian carcinoma and has been extensively employed as a study model due to its resistance to certain chemical compounds.

We found that SKOV3 cells treated for 24 and 48 h, with different concentrations (15, 30, 60, 125, 250, 500, and 750 µg/mL) of the compounds, showed a significant reduction in cell viability in a time- and concentration-dependent manner. However, the most notable effect was observed when the cells were treated with the ND-MET complex, as shown in Figure 5a,b. Some studies indicate that the inhibition of cell viability also depends on the size and shape of the nanoparticles [1,4,6]. It has been reported that ND does not exhibit toxicity for SKOV3 with sizes of 95 nm at a concentration of 10 µM/mL [46].

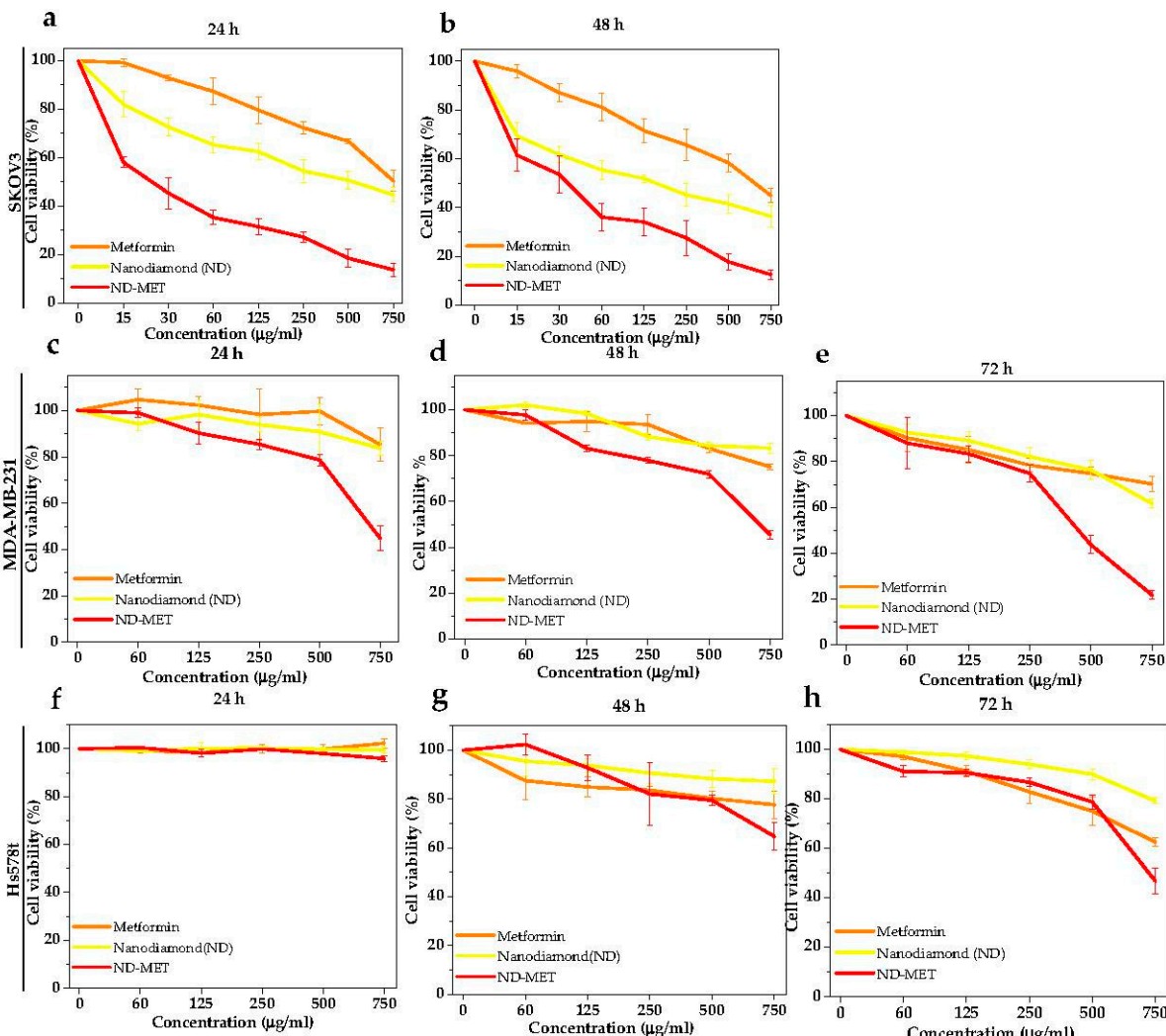

**Figure 5.** (**a**–**h**) Dose-response curves were generated for three cancer cell lines (SKOV3, MDA-MB-231, and Hs578t) and treated with different concentrations of metformin, ND-COOH, and ND-MET for 24, 48, and 72 h. All the values are expressed as average and standard deviation (n = 3).

The half maximal inhibitory concentration ($IC_{50}$) of MET alone at 24 and 48 h was 914 ± 127 µg/mL and 632 ± 116 µg/mL, respectively. The $IC_{50}$ of ND-COOH at 24 and 48 h was 458 ± 139 µg/mL and 157 ± 64 µg/mL, respectively. The ND-MET complex was the one that most effectively decreased the cell viability of ovarian tumor cells since $IC_{50}$ was 24 ± 7 µg/mL and 35 ± 14 µg/mL at 24 and 48 h, respectively (Figure 5a,b; Table 2).

**Table 2.** Half-maximum inhibitory concentration ($IC_{50}$) values for metformin, nanodiamonds (ND-COOH), and nanodiamond-coupled metformin (ND-MET) in three cancer cell lines at 24, 48, and 72 h.

| Cancer Cell Lines | Time | | | | | | | | |
|---|---|---|---|---|---|---|---|---|---|
| | 24 h | | | 48 h | | | 72 h | | |
| | MET | ND-COOH | ND-MET | MET | ND-COOH | ND-MET | MET | ND-COOH | ND-MET |
| SKOV3 | 914 ± 127 | 458 ± 139 | 24 ± 7 ** | 632 ± 116 | 157 ± 64 | 35 ± 14 ** | UD | UD | UD |
| MDA-MB-231 | UD | UD | 842 ± 81 | UD | UD | 801 ± 37 | UD | UD | 454 ± 49 |
| Hs578T | UD | UD | UD | UD | UD | UD | UD | UD | 759 ± 44 |

UD = Undetermined. Statistical analysis was performed using a one-way ANOVA test (** $p < 0.001$).

Preliminary tests that we carried out with the two breast cancer lines (MDA-MB-231 and Hs578t) showed that the effect of the different compounds on these lines is less compared to the ovarian line (SKOV3). Therefore, we decided to carry out the cytotoxicity assays at an additional time point (72 h), and the range of concentrations analyzed was increased from 60 to 750 µg/mL (Figure 5c–h).

As observed in Figure 5c–e and Table 2, only ND-MET resulted in an inhibition percentage greater than 50% for the MDA-MB-231 cell line under the study conditions.

The $IC_{50}$ values calculated for ND-MET in MDA-MB-231 at 24, 48, and 72 h were $842 \pm 81$, $801 \pm 37$, and $454 \pm 49$ µg/mL, respectively. Likewise, in the Hs578t cell line, only the ND-MET complex achieved an inhibition percentage of 50% up to 72 h, with an $IC_{50}$ value of $759 \pm 44$ µg/mL (Figure 5h; Table 2). For the other compounds alone (MET and ND-COOH), 50% cell inhibition was not reached even after incubating with the maximum concentration of the compounds and the maximum exposure time (Figure 5e,h). These analyses show that the cytotoxicity of the different nano-compounds depends on the concentration, treatment time, and cell type.

Finally, the $IC_{50}$ values for ND-MET were analyzed (Table 2) and compared with the $IC_{50}$ of a PLGA-PEG nanoparticle complex loaded with metformin [16], both evaluated in the ovarian cell line SKOV3. We observed that ND-MET presented a 50% inhibition of $35 \pm 14$ µg/mL at 48 h of treatment, while with the PLGA-PEG complex loaded with MET, it was 1295.51 (no deviation standard reported) µg/mL for the same treatment duration. This result shows a high percentage of inhibition of the synthesized complex.

## 4. Conclusions

A simple method to functionalize dispersed nanodiamonds with metformin by direct binding to 1,6-hexanediol was proposed. The cytotoxic effect of nanodiamonds coupled with MET on cancer cell lines is higher than their components. This cytotoxicity is attributed to the effect of metformin and the characteristics of the ND-MET complex. The size of the complex also makes it a biocompatible material, which is an essential factor for various biomedical applications. Furthermore, it is a compound with a homogeneous and monodisperse solubility. This property is because the functionalization makes the 1,6-hexanediol act as an anionic surfactant, which improves its biodistribution and makes it more stable. Additionally, functional groups like carboxyl and amino increase the solubility of MET, which have low bioavailability, thus allowing the molecule to interact electrostatically with the nanodiamond system. Some drugs are efficiently incorporated with this type of electrostatic interaction because the ND-COOH, given its liposolubility, can carry it to the interior of the cell, and the inhibition obtained is attributed to these characteristics. All these factors allow an effective and directed biomolecular interaction, so it is estimated that once evaluated in vivo, it will reduce the side effects.

We found that SKOV3 cancer cells treated for 24 and 48 h with different concentrations of the compounds showed a significant reduction in cell viability in a time- and concentration-dependent manner. However, the most notable effect was when the cells were treated with the ND-MET complex. Some studies indicate that the inhibition of cell viability also depends on the size and shape of the nanoparticles. Unexpectedly, our findings suggest that the carrier exhibits a pronounced effect on SKOV3 cells, surpassing that observed in breast cancer cell lines, which has been reported in different studies; thus we expected to find some toxicity with nanoparticles alone. Therefore, the use in preclinical models such as mouse models will require additional standardization, as the carrier's impact in a clinical setting will depend on a variety of factors including new doses, inoculation method, weight, and immune system status.

**Author Contributions:** Conceptualization, L.E.A.-A. and E.O.-B.; methodology, L.E.A.-A. and A.S.C.-D.; validation, O.N.H.-d.l.C. and G.S.-S.; formal analysis, L.E.A.-A., M.M.-Y. and J.S.S.-L.; investigation, L.E.A.-A., A.S.C.-D., O.N.H.-d.l.C. and G.S.-S.; resources, C.L.-C., E.A.Z.-C. and R.R.-M.; writing—original draft preparation, L.E.A.-A.; writing—review and editing, L.E.A.-A., E.O.-B., M.M.-Y., A.S.C.-D., O.N.H.-d.l.C., J.S.S.-L., C.L.-C. and E.A.Z.-C.; visualization, L.E.A.-A.; supervision, E.O.-B.; project

administration, L.E.A.-A.; funding acquisition, E.O.-B. All authors have read and agreed to the published version of the manuscript.

**Funding:** This research was funded by the Consejo Nacional de Humanidades, Ciencias y Tecnologias (CONAHCYT), grant number 296401.

**Data Availability Statement:** Not applicable.

**Acknowledgments:** This project was carried out at the Centro de Investigación en Materiales Avanzados, S.C., and the Universidad Autónoma de la Ciudad de México. LEAA was a recipient of the 754406 CONAHCYT scholarship.

**Conflicts of Interest:** The authors declare no conflict of interest.

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
