# Peer review of "The Improved Cytotoxic Capacity of Functionalized Nanodiamonds with Metformin in Breast and Ovarian Cancer Cell Lines"

_processes, doi:10.3390/pr11092616_

Round 1

Reviewer 1 Report

In this paper authors have generated a nanodiamond as drug delivery system for enhancing metformin activity in breast cancer and ovarian cancer cell lines. The overall this study and data presentation seems perfect; I will not ask for further experimentations, but at the same time I have few concerns here for authors those I believe require some clarifications in the manuscript in introduction or discussion sections:

1.     NDHEXAMET means the 6 met molecules are attached to one ND-COOH??

2.     As this Metformin is a treatment for type-2 diabetic patients, what would be the inferior impact of this agent when a cancer patient not having type-2 diabetes will consume at higher dose then daily recommendations?

3.     Seems like the drug metformin also have some contaminant of N-nitrosodimethylamine which is a probable cancer-causing agent, and therefore there is some news from FDA about metformin recall (banned). How the authors will comment this situation?

4.     The invitro data seems like ND-COOH as a drug carrier also have some toxicity (near to 40%) in ovarian cancer cell line. Generally, drug carrier should not show the toxicity as such. I am just wondering this carrier at clinical settings may have a significant side effect??

5.     Please mention the significant P-values those are missing throughout of the manuscript.

6.     Figure legend in fig 5 elaborating a, b, c is missing.

Minor English editing's are required.  

Author Response

Comment 1. NDHEXAMET means the 6 met molecules are attached to one ND-COOH?

Answer: No, when the acid-base reaction is carried out, the COOH group changes, the HEXA terminology refers to the hexyl substituent (6-carbon chain that separates the nanodiamond and the MET drug). But due to the confusion, we have decided to change the name of the complex in the manuscript to ND-MET, which refers to the binding of the nanodiamond to metformin.

Comment 2. As this Metformin is a treatment for type-2 diabetic patients, what would be the inferior impact of this agent when a cancer patient not having type-2 diabetes will consume at higher dose then daily recommendations?

Answer: Studies show that the maximum recommended daily dose of Metformin should not exceed 2000- 2500 mg, if the dose is exceeded it has been observed that this drug can generate weight loss, diarrhea, abdominal pain, nausea and in some more severe cases can cause lactic acidosis, however, the use of nanodiamonds would be helping to decrease the dose given of metformin, since our results show that ND-MET has a greater effect compared to free metformin thus minimizing its effects.
In addition, interim analyses of ongoing studies involving metformin treatment of newly diagnosed breast cancer patients have shown that metformin is safe, effective and well tolerated, and exhibits favorable effects on insulin metabolism and tumor cell proliferation and apoptosis [Ritwika Mallik, Tahseen A. Chowdhury,Metformin in cancer,Diabetes Research and Clinical Practice,Volume 143,2018,Pages 409-419,ISSN 0168-8227, https://doi.org/10.1016/j.diabres.2018.05.023]
However, more clinical research is needed to fully appreciate the impact of metformin on cancer recurrence and survival.

Comment 3. Seems like the drug metformin also have some contaminant of N-nitrosodimethylamine which is a probable cancer-causing agent, and therefore there is some news from FDA about metformin recall (banned). How the authors will comment this situation?

Answer: The contaminant mentioned was not used in the synthesis.According to FTIR and RAMAN we did not find sufficient evidence that the mentioned contaminant was present.

Comment 4. The invitro data seems like ND-COOH as a drug carrier also have some toxicity (near to 40%) in ovarian cancer cell line. Generally, drug carrier should not show the toxicity as such. I am just wondering this carrier at clinical settings may have a significant side effect?

Answer: We thanks to reviewer for the wise comments. Indeed, our findings suggest that the carrier exhibits a pronounced effect on SKOV3 cells, surpassing that observed in breast cancer cell lines. This behavior has been previously reported in different studies; thus, it was expected to find some toxicity with nanoparticles alone. Therefore, as you well pointed out its use in preclinical model in mouse model will require additional standardization as the carrier's impact in a clinical setting will depends on a variety of factors including new doses, inoculation via, weight, immune status, etc. In a more complex system, such as the mouse or human bodies, the carrier can interact with a range of molecules that could potentially alter both its effect and toxicity, as well as its metabolism and elimination. Therefore, it is essential to conduct further in vitro experiments, animal testing, and ultimately clinical trials to thoroughly investigate the safety and specificity of the complexes.

Currently, we are working on the optimization in the synthesis of novel NDHEXAMET complexes. Variants are being synthesized with a greater number of metformin molecules attached to NDHEXA-CO2H, aiming to enhance their impact on cellular viability using lower compound concentrations. We hope this approach can mitigate the intrinsic effects of the metformin carrier in ongoing studies.

Finally, we really appreciate the reviewer observations which we will take in account in future in vivo settings. The reply to these limitations has been added in conclusions section.

Comment 5. Please mention the significant P-values those are missing throughout of the manuscript.

Answer A one-way analysis of variance (ANOVA) and a Tukey post-hoc test (Wawruszak, A., et al (2015). Assessment of Interactions between Cisplatin and Two Histone Deacetylase Inhibitors in MCF7, T47D and MDA-MB-231 Human Breast Cancer Cell Lines - An Isobolographic Analysis. PloS one, 10(11), e0143013. doi.org/10.1371/journal.pone.0143013) were conducted to investigate differences in the IC50 across three treatment groups. The one-way ANOVA revealed a significant effect of treatment group on IC50 (p < 0.001), indicating significant differences in IC50 between the treatment groups. Subsequently, a Tukey post-hoc test was performed to compare the mean IC50 across the treatment groups. The Tukey test results demonstrated a significant difference in IC50 between the "ND 24 h" and "ND-Met 24 h" groups (p<0.001). Specifically, the "ND-Met 24 h" group had a significantly lower IC50 compared to the "ND 24 h" group. Missing P values have been added to figure 5 legend, a supplementary table and indicated in materials and methods.

Materials and methods:

For IC50 data, a one-way analysis of variance (ANOVA) and a Tukey post-hoc test were conducted to investigate differences in the IC50 across three treatment groups.

Comment 6. Figure legend in fig 5 elaborating a, b, c is missing

Answer: As you correctly pointed out, the numbering had not been placed in Figure 5, so it was added in the manuscript directly.

Reviewer 2 Report

1.     The description and the figure are not inconsistent, which make reviewer confused. In line 189, b indicates NDHEXA-CO2H; In figure 1, b indicates 1N NaOH, KMnO4 and HCl; In figure 2, b indicates NDHEXA. Please revise.

2.     In cytotoxicity assay, the concentration gradient setting of NDHEXAMET nanoparticle is unreasonable, so the IC50 values and the conclusion are not convincing.

3.     In line 376-378 “This is attributed to the effect of metformin and the characteristics of the NDHEXAMET complex, such as a radially symmetrical shape and the dimension of 5 nm since with this size it has more surface area and the ability to permeate the cell.” The author should conduct the cell uptake experiment to support this conclusion.

4.     In quantification result, there are no significant differences in figures. Please add.

Some sentences make me confused. Need to improve.

Author Response

Comment 1. The description and the figure are not inconsistent, which make reviewer confused. In line 189, b indicates NDHEXA-CO2H; In figure 1, b indicates 1N NaOH, KMnO4 and HCl; In figure 2, b indicates NDHEXA. Please revise.

Answer: As you correctly pointed out, there was a confusion in indicating that NDHEXA-CO2H corresponds to b in Figure 1, b shows what was added to form NDHEXA-CO2H which was 1N NaOH, KMnO4 and HCl.

On the other hand b in Figure 2 corresponds to the infrared spectrum of NDHEXA, which is independent of Figure 1, so no modification was made here.

Comment 2. In cytotoxicity assay, the concentration gradient setting of NDHEXAMET nanoparticle is unreasonable, so the IC50 values and the conclusion are not convincing

Answer: We appreciate your wise and assertive comments that greatly enrich our manuscript. We apologize for the mistakes in the drawing of figure 5. Now, we have thoroughly revisited the plots pertaining to the cytotoxicity assays and corrected the gradient configurations accordingly, as we detected an error in the axis, as you pointed out. Furthermore, we have incorporated one-way analysis of variance (ANOVA) and post-hoc Tukey's test into our analysis to investigate differences in the IC50 across the three treatment groups. Figure 5 have been corrected, and missing P-values added (Wawruszak, A., et al (2015). Assessment of Interactions between Cisplatin and Two Histone Deacetylase Inhibitors in MCF7, T47D and MDA-MB-231 Human Breast Cancer Cell Lines - An Isobolographic Analysis. PloS one, 10(11), e0143013. doi.org/10.1371/journal.pone.0143013)

Comment 3. In line 376-378 “This is attributed to the effect of metformin and the characteristics of the NDHEXAMET complex, such as a radially symmetrical shape and the dimension of 5 nm since with this size it has more surface area and the ability to permeate the cell.” The author should conduct the cell uptake experiment to support this conclusion.

Answer: Given the characteristics of the nanodiamond to permeate the cells due to its nanometer size, it could be assumed that it enters the cell, however, the reviewer is right, so we have decided not to assume such conclusion and change the conclusion because the requested cellular uptake assay requires material and time that we do not have to carry it out.

Comment 4. In quantification result, there are no significant differences in figures. Please add.

Answer: If the reviewer refers to the COOH site quantification results in Table 1, the standard deviation data was added to indicate significant figures.

In response to your suggestion, with the Grammarly program, to which one of the authors is subscribed, we made the necessary corrections so that the English goes in the best grammatical form.

Reviewer 3 Report

The paper does not present enough evidence to prove their claim. Authors have used 3 different cell lines and does not provide any rationale for using so. They have concluded the effect of nanodiamond met against ovarian and breast cancer cell lines while they have not testes any other cancerous cell lines to show the specificity. Similarly, authors have not used a negative control (normal cells) as well.

Author Response

Comment 1: Authors have used 3 different cell lines and does not provide any rationale for using so.

Answer:

We appreciate your wise comments and questions. As you correctly pointed out, it is necessary to provide a justification in the manuscript for the use of the 3 cell lines in our assays.

The rationale for the use of these well studied cancer cell lines have been documented in so many publications. In breast cancer, the subtype that typically exhibits a lower response to therapies and has a poorer prognosis is triple-negative breast cancer (TNBC) [(Bai, X., Ni, J., Beretov, J., Graham, P., & Li, Y. (2021). Triple-negative breast cancer therapeutic resistance: Where is the Achilles' heel?. Cancer letters, 497, 100–111. https://doi.org/10.1016/j.canlet.2020.10.016)].The lack of estrogen, progesterone, and HER2/neu receptors in TNBC renders this type of cancer unresponsive to hormone-targeted therapies, such as tamoxifen, which are effective in tumors expressing estrogen or progesterone receptors. Moreover, due to the lack of HER2/neu receptor expression, TNBC does not respond to HER2-targeted therapies, like trastuzumab[(Dawson, S. J., Provenzano, E., & Caldas, C. (2009). Triple negative breast cancers: clinical and prognostic implications. European journal of cancer (Oxford, England : 1990), 45 Suppl 1, 27–40. https://doi.org/10.1016/S0959-8049(09)70013-9; Waks, A. G., & Winer, E. P. (2019). Breast Cancer Treatment: A Review. JAMA, 321(3), 288–300. https://doi.org/10.1001/jama.2018.19323; Bianchini, G., Balko, J. M., Mayer, I. A., Sanders, M. E., & Gianni, L. (2016). Triple-negative breast cancer: challenges and opportunities of a heterogeneous disease. Nature reviews. Clinical oncology, 13(11), 674–690. https://doi.org/10.1038/nrclinonc.2016.66)]. Consequently, the inefficacy of these specific therapeutic options makes TNBC more challenging to treat when compared to other subtypes of breast cancer. Hence, we decided to evaluate the effect of nano-compounds as novel therapeutic tool.

In ovarian cancer (OC), the serous subtype is highly prevalent and can exhibit high aggressiveness, as well as being associated with a poor prognosis and reduced responsiveness to therapies, compared to other OC subtypes. SKOV3 is a cell line derived from serous ovarian carcinoma and has been extensively employed as a study model due to its resistance to certain chemical compounds traditionally used for OC treatment.

Therefore, the use of these 3 cell lines aims to enhance our biological knowledge concerning different types of cancer that are difficult to treat and are associated with a poor prognosis, in addition to exhibiting a limited response to standard therapies. The knowledge generated could aid in proposing novel therapeutic alternatives based on nano-compounds that overcome this therapeutic resistance.On the other hand, these 3 cell lines have been extensively studied and characterized in cancer research, simplifying their handling and the comparison of results with other studies. Additionally, their ease of cultivation under standard conditions makes them suitable for our assays.

These comments and explanations have been added to the revised version of manuscript (Page 10, lanes 447-461).

Comment 2: They have concluded the effect of nanodiamond met against ovarian and breast cancer cell lines while they have not testes any other cancerous cell lines to show the specificity.

Answer: To assess the effect of nanocompounds on tumor cells, we initially focused on the ovarian cancer cell line SKOV3. The preliminary results indicated a cytotoxic effect of the nanocompounds on the SKOV3 line, which led us to investigate cells with different origin. Thus, we selected two triple-negative breast cancer (TNBC) cell lines for further evaluation. These experiments revealed that the nanocompounds also exhibited an effect on the TNBC cell lines, albeit to a lesser extent compared to the ovarian line. This suggests that the cytotoxic impact of the nanocompounds may not be limited to a specific cancer type. To gain more comprehensive insights, in the future, cytotoxicity assays on cell lines with diverse characteristics will be necessary to elucidate the full impact of the nanocompounds. Nonetheless, these findings highlight the potential of nanocompounds in reducing the viability of various cell types, offering a promising avenue for the treatment of patients with diverse types of cancer.

Comment 3: Similarly, authors have not used a negative control (normal cells) as well.

Answer: As you correctly pointed out, we have not evaluated non-tumorous cell lines; however, their absence does not invalidate our research, as we focused here in the study of its effects in cancer cells. It is indeed crucial to assess these "normal" cell lines with nanocompounds to determine their cytotoxic effects and identify potential risks associated with the therapy, however the best way to achieve this is using animal a model to evaluate potential damage to tissues and organs. This approach will help us in developing targeted therapy that selectively attacks tumor cells while sparing normal cells or having minimal impact on them. We acknowledge the importance of analyzing the effect of nanocompounds on non-tumorous cell lines to establish their safety and specificity in vivo.

We are aware that these cell lines may provide limited information. To gain more comprehensive insights and identify potential risks before advancing to a clinical phase, the use of more complex models, such as animal experimentation, becomes necessary. Lastly, we sincerely appreciate your valuable observations, and some of these comments have been incorporated into the manuscript to enhance its clarity, and serves us as an important guide for future studies.

Round 2

Reviewer 3 Report

The paper can be accepted in the present form.

Author Response

Dear reviewer, thank you for your suggestion that the article be accepted in its present form, the editor has recommended us to revise the format of table 2 to make it the same as table 1, which we have carried out.